# Knowledge Management for Sustainable Development in the Era of Continuously Accelerating Technological Revolutions: A Framework and Models

**Meir Russ** 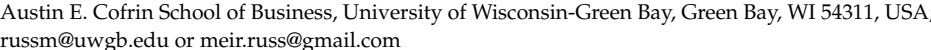

Austin E. Cofrin School of Business, University of Wisconsin-Green Bay, Green Bay, WI 54311, USA;
russm@uwgb.edu or meir.russ@gmail.com

**Abstract:** This conceptual, interdisciplinary paper will start by introducing the commencement of a new era in which human society faces continuously accelerating technological revolutions, named the Post Accelerating Data and Knowledge Online Society, or '*Padkos*' ("food for the journey; prog; provisions for journey"—in Afrikaans) for short. In this context, a conceptual model of sustainable development with a focus on knowledge management and sharing will be proposed. The construct of knowledge management will be unpacked into a new three-layer model with a focus on the knowledge-human and data-machine spheres. Then, each sphere will be discussed with concentration on the learning and decision- making processes, the digital supporting systems and the human actors' aspects. Moreover, the recombination of new knowledge development and contemporary knowledge management into one amalgamated construct will be proposed. The holistic conceptual model of knowledge management for sustainable development is comprised by time, cybersecurity and two alternative humanistic paradigms (*Homo Technologicus* and *Homo Sustainabiliticus*). Two additional particular models are discussed in depth. First, a recently proposed model of quantum organizational decision-making is elaborated. Next, a boundary management and learning process is deliberated. The paper ends with a number of propositions and several implications for the future based on the deliberations in the paper and the models discussed and with conclusions.

**Keywords:** *Padkos*; continuous technological revolutions; sustainable development; knowledge sharing; knowledge management; *Homo Technologicus*; *Homo Sustainabiliticus*; conceptual model; quantum organizational decision-making

## 1. Introduction

Technology luminaries and economists are suggesting that we are amid the 4th industrial revolution (see example at [1]), and we will be possibly facing the fifth society in the near future [2]. Alternatively, economists and sociologists have suggested different taxonomies of economic waves for such times, such as Kondratieff's waves (see example at [3]), or cyclic crisis [4]. Social scientists have pointed to wars, social or political revolutions, or pandemics as social mechanisms that have allowed human society to move from one technological 'pick' (or economic 'valley') to another. This paper is suggesting terminating the counting of revolutions and/or cycles since we are now (and for the foreseeable future) facing a continuous, ongoing, simultaneous and accelerating wave of technological revolutions, many of which are 'general purpose technologies'. As such, an ongoing tsunami that is accelerating is a better metaphor (or an infliction/tipping point [5]) for the distinctive and perilous time like this in the history of the human species on this planet.

The past industrial revolutions or economic cycles were all ascertained with the prism of the tenure of human life. Revolutions happened ever so often, and when they occurred, it happened once within the life span of an individual. Following such revolutions, individuals and societies had adequate time to adjust. This paper is proposing (as a conjecture)

that at present, such a luxury is no longer available. There are no times in betweenness that will allow the individual and/or society to adjust, and that if humanity wants to survive as a species (which is already questionable-see the Sustainabilism paper [6]), as individuals and as a humanistic society, there is a need to develop a different new mechanism. Such a society is defined here as the Post Accelerating Data and Knowledge Online Society, or '*Padkos*' ("food for the journey; prog; provisions for journey"—in Afrikaans) for short. The implications listed at the end of this paper are intended to be a 'food for thought' for the reader for the journey ahead. In a future paper such a process will be proposed, one that could facilitate such an expeditious transition if this infliction point has not been crossed yet.

This conceptual paper has five novel contributions. First, the paper will establish that humanity is facing a challenge not yet faced in the past by *Homo Sapiens*, as individuals and as societies, and that learning from the past might be of little relevance for finding a livable solution. Next, the paper will pursue the development of a model of knowledge management for sustainable development within this unique context, while considering the most recent developments of big data, machine learning and transitioning into the age of a circular and smart sustainable economy, all in the context of cybersecurity, humanistic paradigms and time. Moreover, the paper is proposing a three-layer model of knowledge management (KM) that includes a human layer, and a machine layer and a unified new knowledge development and knowledge sharing layer. Additionally, the paper suggests combining decision making and learning into a unified, synergistic managerial activity. Finally, the paper proposes a boundary management and learning model specific to knowledge sharing.

The remainder of the paper is organized into the following sections. In Section 1.1 the author stipulates the beginning of the *Padkos* era by recognizing the different aspects of the discontinuous change the economy and society are going through since the 1980s. In Section 1.2, the methodology used in developing the different models, as described later, is discussed. Section 2 encompasses most of this paper and covers the models of KM for sustainable development as follows: Section 2.1 describes the top layer of the meta-system layer model, namely the sustainable development model with KM embedded with its five components. Section 2.2 discusses the three layers of the KM mesosystem model, which are detailed in Sections 2.2.1, 2.2.3 and 2.2.4. Section 2.2.1 describes the human/knowledge layer while Section 2.2.4 describes the machine/data layer. Section 2.2.3 discusses the simultaneous (Yin-Yang) model of new knowledge development and contemporary knowledge sharing. Section 2.2.2 is discussing the dimension of time which plays a critical role in all models described in this paper, but most importantly when discussing the "Quantum Model of Decision-Making" which is covered both in Sections 2.2.1.2 and 2.2.4.2. Sections 2.2.1.1 and 2.2.4.1 describe the learning models in the knowledge and data layers. Sections 2.2.1.3 and 2.2.4.3 describe the digital system aspects in the knowledge and data layers. Sections 2.2.1.3 and 2.2.4.4 discuss the role of the human actors in the knowledge and data layers. Next, the paper covers the ethical aspect (in Section 2.2.5), discussing in detail the *Homo Technologicus* and *Homo Sustainabiliticus* paradigms; and in Section 2.2.6 the cybersecurity aspect as relevant to this paper are covered. To conclude Section 2, the model of boundary management and learning is discussed in Section 2.3. Seven propositions are advanced toward the end of Section 2. The paper closes with Section 3, where the ensuing mega-trends are discussed in Section 3.1. Then, in Section 3.2 the implications for research-theory building and model testing are discussed. Next, the implications for policy making (Section 3.3) and for practitioners (Section 3.4) are reviewed. The paper closes with conclusions in Section 4.

### 1.1. The Current State

Time between major revolutions is shrinking exponentially. On a logarithmic scale, the ratio between 'the time to next paradigm shift' and between 'time before present' is about 1:1 [7], while life expectancy worldwide went from approximately 35 years old in

1920, to 46 in 1950, to about 70 in 2010 [8–10]. The figure below (Figure 1) is a simplistic, schematic graph, suggesting that somewhere around 1970–1980 the two lines crossed, meaning that after that point in time, throughout one's lifetime there was more than one major technological paradigm shift, and that since about 10 years ago, one would experience multiple paradigms shifts during their life-time, leaving practically no time for society to adjust. For example, researchers identify six (6) major technological paradigms developed between 1990 and today, just in personal computing devices [11].

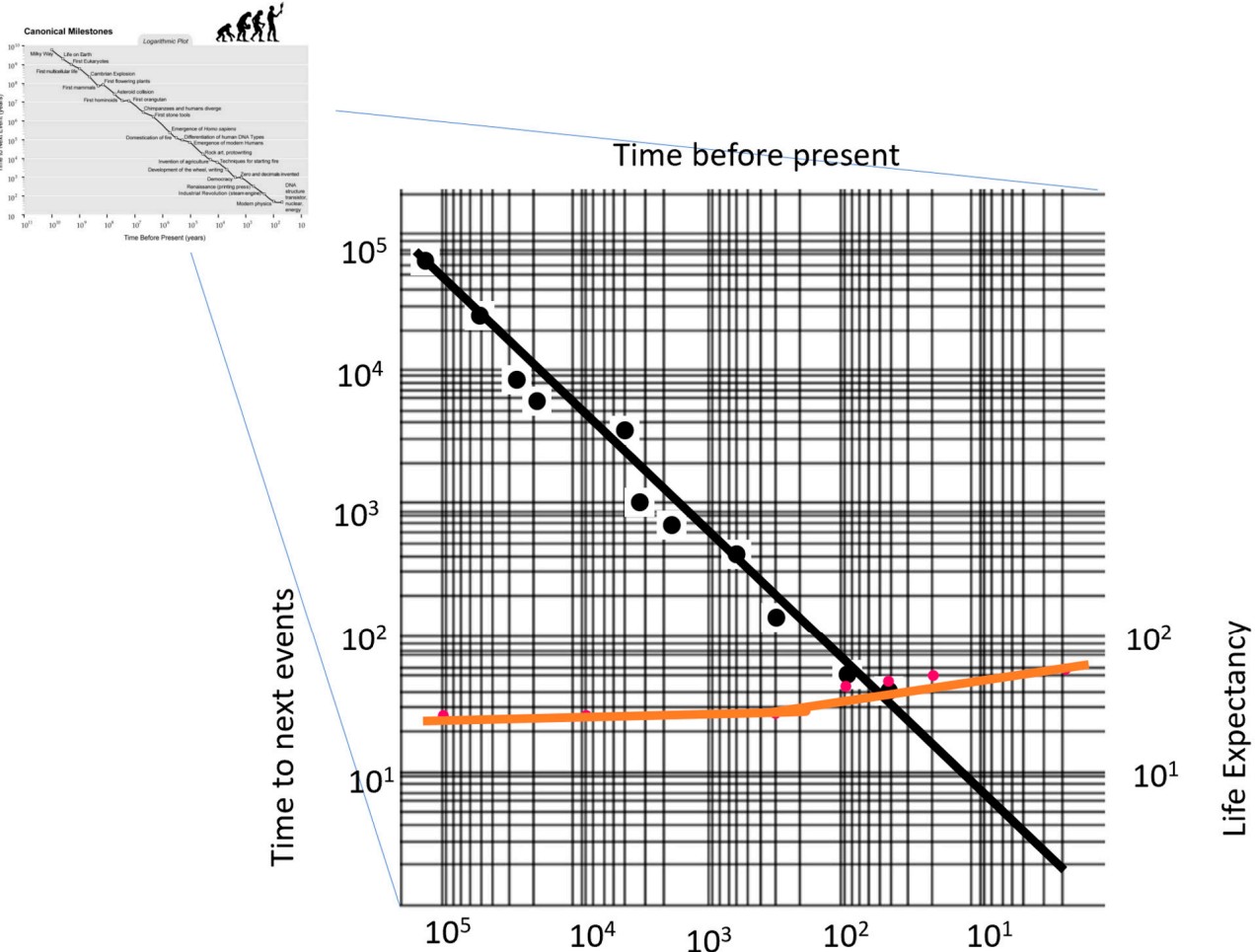

**Figure 1.** 'The time to the next paradigm shift' and Life Expectancy. Source: Author's elaboration on Singularity [7].

Biological evolution pace has been overtaken by memetic evolution pace (e.g., we are able to edit genes [12]; conduct experiments in the newly emerging field of mechanobiology [13]) and in the very near future, face the situation where the pace of change in the environment will be faster than the processing speed of the human brain, while machine intelligence is turning out to be 'smarter' than human intelligence and probably not controllable [14]. All this and more is resulting in a 'singularity', becoming a physical and biological reality and not the 'stuff' of science fiction or philosophical discussions [15]. Moreover, the half-life of knowledge, estimated currently to be about 18–24 months in medicine [16], high-tech, information systems, and in management, is expected to be less than 12 months in 2050 [17,18].

The merging of the two trends: the continuous accelerated pace of general-purpose technological revolutions and the 'evolution' moving the neural-data space from the biological space to the combined DNA, neural, digital space, caused numerous scientific, technology development and investment advances. Chief among the scientific and technology development are the preparations for the 'singularity,' meaning enhancing human brain capacity

and preparing for the merger of man and machine, what had been described as *Homo Technologicus* [6,19]. For example: (1) research in communication between brains [20,21]; (2) research in memory manipulation, including erasing and/or writing memories [22]; (3) developing technology that can facilitate communication between the human brain and external devices such as artificial limbs [23], computers [24] or a cellphone [25].

In parallel, there are also unique developments in the economic financial sector. For example, central bank policies' and new financial options, globalization of the financial markets while using the latest communication technologies, and lack of legislation [26] are creating increasingly concentrated financial national and global markets with a higher risk of crashing. Research by Deutsche Bank found that in the last 400 years, the frequency of financial crisis significantly increased since the early 20th century [27].

Another imperative response to the technological mega trends was the recognition of importance of intangibles—during the early 1990s [28] and the documented increase in investments in intangibles (software, knowledge, human capital). In the mid-1990s (in the US) the private investment in (measurable) intangible capital passed the investment in tangible capital [29], and in 2017, the rate of investment in intangibles almost doubled the investment in tangibles [30]. The OECD estimate that the total investment in intangibles is about twice the reported investment [31] since we still do not have an adequate accounting and financial system to capture the majority of the investment in intangibles, so significant portions of such investments are still hidden [32].

The growing importance of intangibles as a resource and the growth of the service/intangible markets, the concentration of power and capital in the financial markets and the pervasiveness of communication and digital networks, all gave rise to 'winner takes all' monopolistic market dynamics and companies [33], or what some call superstar companies [34].

As a result, economic inequality in many developed countries (primarily English speaking) is today at the levels not seen since the last depression, about a century ago. The 'Inequality Possibility Frontier' is approaching new levels [35] that are reminders of the inequality documented in Northern Italy and in Holland during the middle ages and early Renaissance [36]. The wealth accumulation growth of the top 0.1% is the highest since such data was collected [35]. About 10% of the world's population lives on less than $1.90 a day [37], while the link between productivity and compensation was practically broken in the early 1980s resulting in stagnations (at best) of compensation (or wages) for the majority of the labor force [38]. Moreover, Ferruzza et al. [39] suggested that the United Nations' sustainable economic development goals (SDG 8; [40]) document the need for a new model for sustainable development combining economic needs with social concerns.

Finally, research is suggesting that driven by the economic and social trends, populism is now seen at a very high level, similar to the one observed prior to the start of World War II [27].

This brief review will be amiss for not mentioning the impact of such an economic development and the pressures of demographics (the anthropogenic-driven regime) on the natural environment [41] driving global warming, climate change [42,43] and the 6th extinction [44], all resulting in major immigration waves [45], and the degradation of quality of live [46] and life expectancy [47].

Finally, the current COVID-19 pandemic is only accelerating the trends listed above [48] and the need for a new model of sustainable development [49].

*1.2. Methodology*

This paper's intention was to broaden the scope and to incorporate the theoretical and empirical implication of the most recent technological advancement concerning knowledge management (KM) in the context of sustainable development as an updated conceptual model. A conceptual model was defined by Miles and Huberman [50] as a visual or written document that explains in graphical and/or in narrative form the key concepts, factors, and variables that encompass it. To accomplish that, the author

started with the most recent model of KM for sustainable systems [51], while adding the summary of the narrative of multidisciplinary literature reviews regarding the evolving labor markets [52] and the new-networked knowledge-based global economy [6]. To frame the updated state of the art causal literature review (a systematic review of such literature is beyond the scope of this paper), Heidegger's [53] ontology of technology and knowledge was used since it allowed for covering the broad scope of the multidisciplinary nature of the subject. The five components of the framework—operational, collaborative, organizational, instrumental, and holistic—were adopted to guide the literature review. This multidisciplinary literature review draws on multiple academic literature. Among them are the labor's economic literature, technology management and business innovation literature, human capital and knowledge management literature, psychological and organizational behavior literature; computer and data science; and machine learning literature. The broad scope of the casual multidisciplinary literature review used was a modified version of an integrative literature review [54] (pp. 356–357). This methodology allows for initial conceptualization resulting in a new model and/or framework, offering a new perspective on a multidisciplinary topic. For example, reviewing the literature of artificial intelligence, machine learning and new knowledge creation (see below in Section 2.2.4.1) while using the modified version of an integrative literature review [54] strongly suggested the addition of an additional layer for knowledge management, one that is concentrated on data and digital machines [55,56]. Using the five components of Heidegger's [53] ontology, suggested to include in this data layer the same building blocks (learning, decision making, systems, human actors) as in the earlier model of KM [51], while questioning the separation between knowledge sharing and new knowledge development. The very recent literature discussing the need for a new model of digital transformation resulting from the need for resilience and innovation "to deal with" the COVID-19 pandemic [57] suggested that the two aspects interact synergistically as incorporated in the middle layer of the model (see Section 2.2.3).

As a result of applying this research procedure and inductive reasoning, a significantly modified multilayer, complex and dynamic conceptual model was assembled, that will require additional foremost verifications in the future.

Next, the updated model for Knowledge Management for Sustainable Development based on and modified from the model presented in [51] will be presented.

## 2. The *Padkos* Model of Knowledge Management for Sustainable Developments

The original model of knowledge management was developed while the author prepared for the editing of the book "Handbook of Knowledge Management for Sustainable Water Systems" [51] which focused on knowledge management in the context of sustainable water systems. With the focus on Knowledge Management, the updated model described here will include three major layers: The Meta-Systems level, The Knowledge Management (KM) mesosystem of major components level, and the detailed KM microsystems with its sub-components level.

### 2.1. The Meta-System Layers

At the meta-system top level, the model includes the Human Systems: political, economic, technological, and social (see discussions and indicators in, for example [58,59], but mostly based on [60], the Natural and Engineered Natural Systems (see example [61]) and the 'Co-evolution' of those two systems (see Figure 2). Such co-evolution results from the impact human activities have (mediated by technology) on the natural and engineered systems and the responses and outcomes of the natural and engineered systems to these activities on the human systems. To make the co-evolution aspect more concrete and sustainable, the circular economy model [62] was added to the economic sphere (see below).

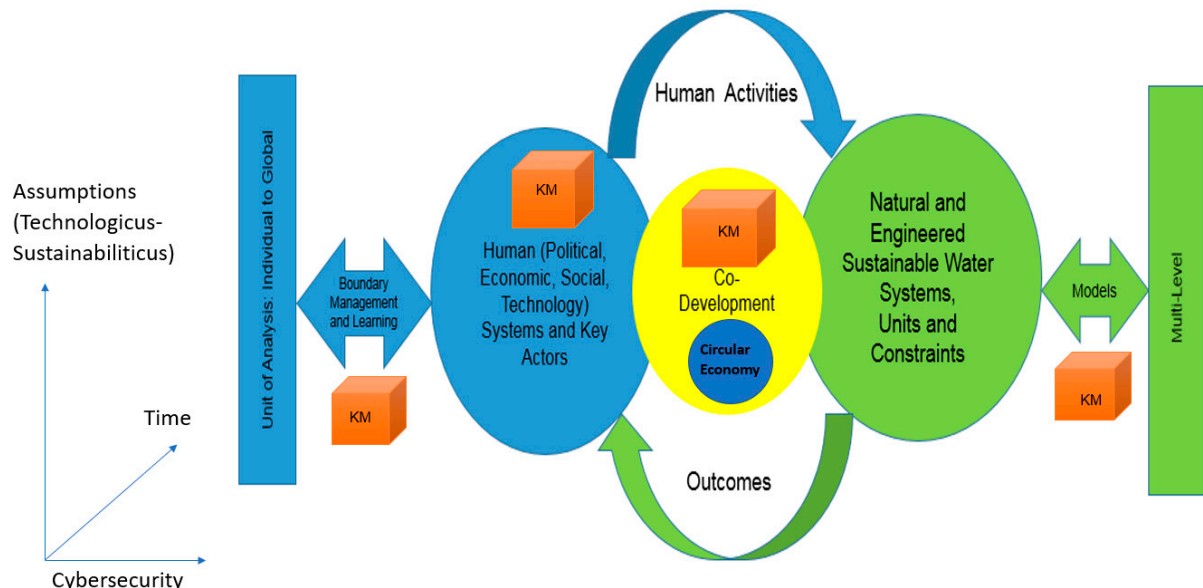

**Figure 2.** Knowledge Management for Sustainable Development. Source: Author's slight modification from Russ [51].

This co-evolutionary model was enhanced by adding on the human system side the different actors or optional units of analysis (as subsystems) that might be relevant to the issue at hand; and on the natural system side, the different levels of the systems that might be relevant to the issue at hand (as subsystems).

On the human system side, we must take into consideration the complexity of the different potential units of analysis involved as building blocks, starting with an individual, teams, organizations and then going up in complexity to inter-organization, national, regional and global units and scales. Each unit has its own learning and decision-making complexity and additional complex sub-units, issues, and boundary management aspects (see excellent discussions of the importance of this complex management system in [63] and potential barriers for KM in [64]). Obviously, there is also an interaction between and within the subsystems (e.g., economic and social or legal) and the subunits (e.g., conflicts of interest between different functions within an organization regarding the adaptation of a new technology). A complex system with multiple time delays and positive and negative feedback, resulting in an unexpected and counter intuitive result should only be expected (see early examples and treatment in [65], which is typical for 'wicked problems' [66,67]).

For the purpose of this model, economic, social and political aspects will focus on sustainable development, which has been defined during the UN Sustainable Development Summit in September 2015 by world leaders as "a plan of action for people, planet and prosperity" devised to "shift the world onto a sustainable and resilient path" [68] (p. 1). The agenda, consistent with earlier definitions of sustainability and Triple Bottom Line [69,70] (as well as this model) have three segments: social, environmental, and economic, including seventeen Sustainable Development Goals (SDGs). In this model, these goals should be framed by the circular economy model (mentioned above) for consistency purposes.

On the sustainable natural system side, the different levels of the systems were added (e.g., in case of water systems, household, city, river basins, etc.). Here, again, similar to the human side, we have multiple and complex interactions within and between different subsystems (e.g., Hydraulic, Atmosphere, Land) and different subunits, which as mentioned above results in a complex system. Each one of them is connected within this framework by models that are used and/or understood by the human actors (see the interesting discussion [71] about two models: stylized and comprehensive). It is critically important to understand that the human side acts and operates on the natural side based on the approximations that the models provide (see an example of a recently proposed

model of Panarchy by Garmestani et al. [72], similar to the actions and results from the coevolution as described above.

To complete this level of the model, Knowledge Management (KM) was added at the heart of every point in the model where human and/or machine learning and decision making takes place (see Figure 2). Obviously, all KM activities are connected, related, and intertwined, including boundary management arrangement [73].

### 2.2. Knowledge Management Mesosystem Model

Delving deeper into the second level of the model is the next step, specifically into the construct of Knowledge Management (KM) for its major components (see Figure 3). Here, the model developed by Russ et al. [74] is used, but significantly modified, with focus on the actors, (or talent), the process, or specifically the learning and decision-making aspects, and the systems, or in this case the knowledge-based systems and the data/artificial intelligence (AI) systems in the context of up-and-coming Digital Native Enterprise [67,75]. As indicated earlier, the pace and the scope of the development of digital technologies is extraordinary, as well as the transition from knowledge being the strategic commodity to data becoming the pervasive and strategic commodity [75] requires the modification of the earlier model [51]. The need for an updated model of KM resulting from the shifting digital transformation [57], the shifting practices in the digital workplace [76], the growing skills gap [77], the new learning opportunities [78], the need to secure KM [79] was also advocated by Värk and Reino [80] that suggested to incorporate personal practice ecology of KM into the workplace.

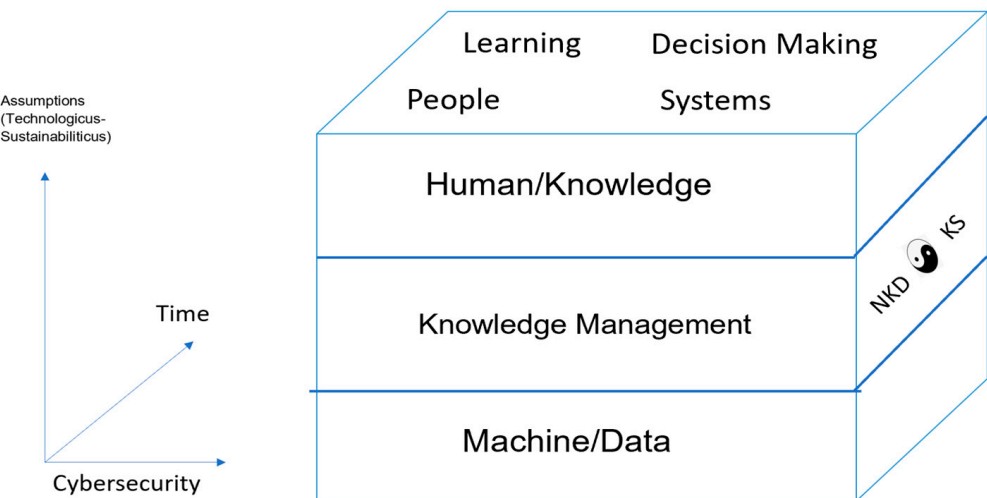

**Figure 3.** The Mesosystem Model of Knowledge Management. Source: Author's elaboration.

The implications for this model are three-fold. First, in place of a single layer model of knowledge management (see Figures 2 and 3 on pages 5, 6 in [51]), this mesosystem model is proposing to include a knowledge layer, a data layer and in between those two, a new knowledge (developed) and existing/present knowledge (to be shared) layer (see Figure 3).

Second, at the process aspect of the model, decision making is added now as an integral part of knowledge management. Intuitively it is obvious that at the present juncture, the planning and thinking stage, data collection, the actual decision and the implementation, and the learning are almost instantaneous, and all aspects must be considered holistically and simultaneously, and they coexist, share and codevelop with what was identified by the literature as knowledge management (see example at [75,81,82]). Lastly, the third modification is the Yin-Yang intertwined and synergistic nature of new knowledge developed and shared almost instantaneously. The most contemporary example and illustration of such a process is the current development of the vaccines for COVID-19 [83].

Examining more in depth this mesosystem model, the three layers of the model are proposed, top down in Figure 3: (a) Knowledge/Human; (b) the Yin-Yang knowledge development and sharing, and the (c) Data/Digital Machine will be briefly explained and are broken down (illustrated) into (by) their specific models (see Figure 4a–d, Figure 5, Figure 6a–d), where each one of the ten building blocks (constructs/concepts) will be briefly discussed. The dimension of time and the model of organizational decision making, due to its importance, will be discussed more in depth as well (see Figure 4b).

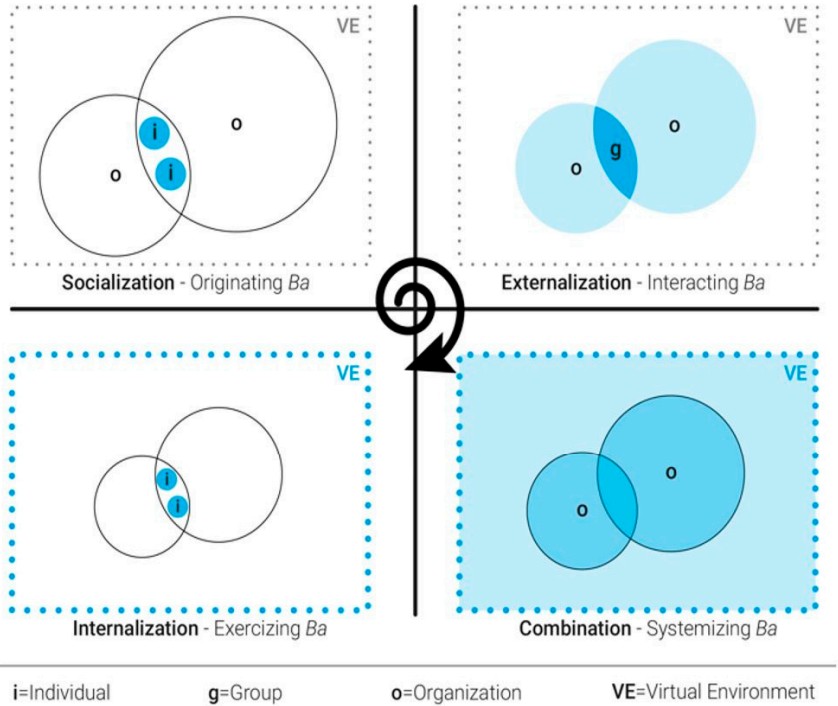

*The four phases of Nonaka's SECI model, adapted to fit the inter-organizational ontological level within a virtual environment.* Source: Niccolini et al., [86].

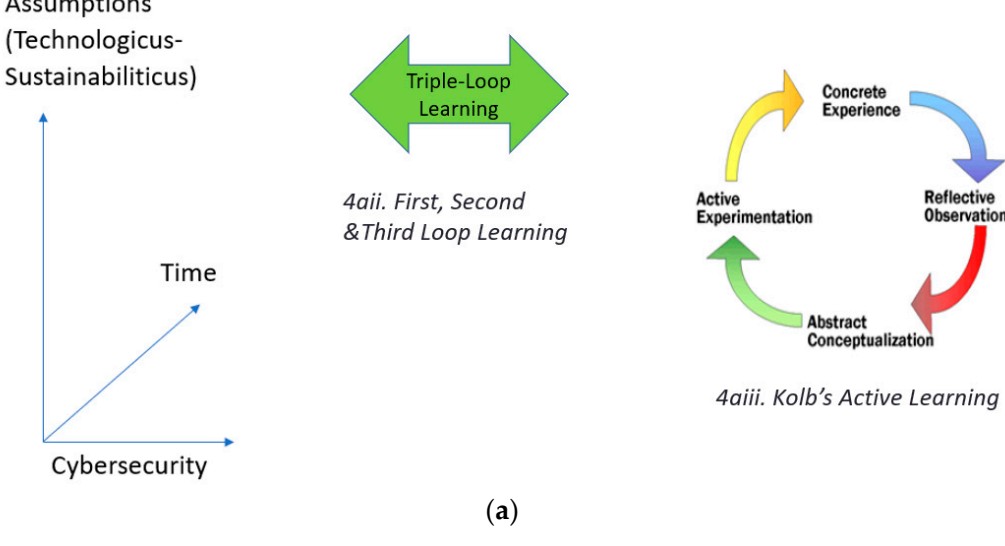

(a)

**Figure 4.** *Cont.*

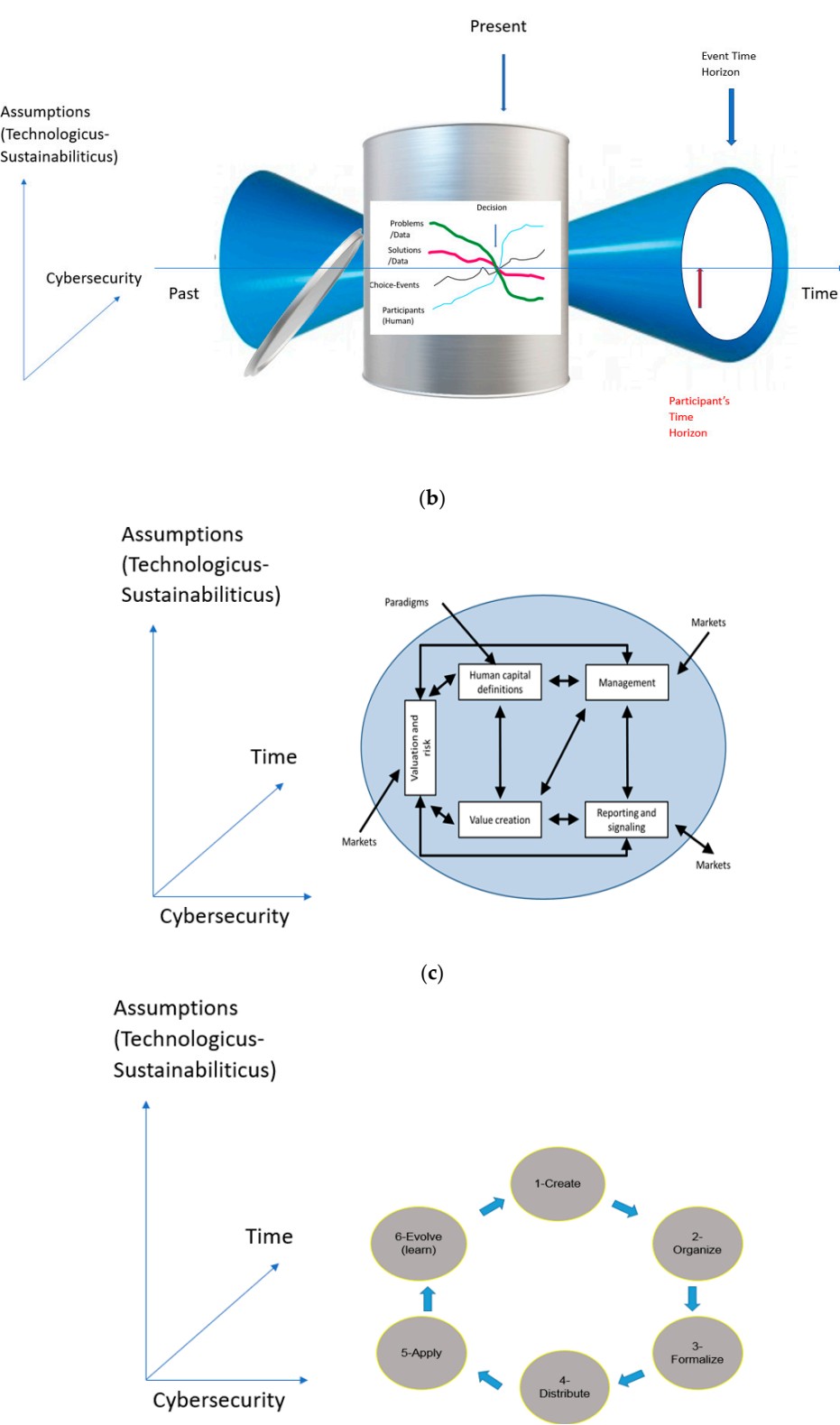

**Figure 4.** (**a**) Learning at the Knowledge/Human layer of Knowledge Management. Source: Author's slight modification of Russ [51]. (**b**) Quantum Organizational Decision-Making at the Knowledge/Human layer of Knowledge Management: The time aspect of quantum organizational decision-making model. Source: Author's slight modification of Russ [51]. (**c**) Human Capital at the Knowledge/Human layer of Knowledge Management. Source: Author's slight modification of Russ [51]. (**d**) Knowledge Based Systems (KBS) at the Knowledge/Human layer of Knowledge Management. Source: Author's slight modification of Russ [51].

An illustration of this model can be seen for example in the information support of web community users' personal verification systems, proposed by Korobiichuk et al. [84]. The proposal incorporates the user and their knowledge at the knowledge/human layer with the data and the verification system (including the cybersecurity aspects listed below in Section 2.2.6) at the data/digital machine layer, for the purpose of knowledge sharing by a web community, at the knowledge sharing layer.

### 2.2.1. Knowledge/Human Layer

First, the four aspects of this layer will be covered: (1) Learning, (2) Decision Making, (3) Human actors, and (4) Knowledge-Based Systems. The discussion regarding Time, Time Horizon and its importance in the organizational decision-making process will be added as well.

#### 2.2.1.1. Learning

For a start at the knowledge layer, the human learning (see Figure 4a) might focus on tacit knowledge using Kolb's [85] active learning model (Figure 4a, or on codified learning using the virtual Ba model illustrated by Niccolini et al. [86] (Figure 4a), or any mix of the two; or others as appropriate for the case; all (potentially) exercising the three feedback loops of learning (e.g., Argyris' double-loop learning [87]); or the review in Tosey et al. of triple loop learning [88] (Figure 4a).

#### 2.2.1.2. Decision Making

The decision-making aspect of this model is the Individual and the Organizational "Quantum Model of Decision-Making" (see Figure 4b) proposed in Russ ([51], Figure 4, p. 7). Some of the typical individual models of decision making discussed in the academic literature include: Utility based models and theories, Rational economics and Irrationality and Behavioral Economics, Framing and Reference Dependence, Bounded Rationality and Decision Heuristics, among others (see examples and reviews in [89]: Six steps in the managerial decision-making process, in Exhibit 9.3, p. 282; Personal decision frameworks in Exhibit 9.5, p. 287; and in [90–92]). Organizational models of decision making include, but not limited to, are administrative [93], search, ad hoc problem solving, exception management [94], and the garbage can model of decision making [95]. The garbage can model of decision making was chosen (for this paper) since it explicitly enables incorporating streams of data and digital information. A more detailed discussion about the aspects of time and time horizon will follow below.

#### 2.2.1.3. Human Actors and Knowledge-Based Systems

Next, the human actors' talent was modeled (see Figure 4c) using the HC praxis model (Russ [96]) and the Knowledge-Based-Systems (KBS) was modeled (see Figure 4d) using the six life cycle stages of KBS (e.g., Russ et al. [97]), including the sustainability aspect of the KBS as well as consideration [98].

### 2.2.2. Time

The truth of the matter is that one aspect is mostly missing from these models and must be explicitly added to the proposed framework, thus adding another element, time (see Figure 4b). Again, and again, while teaching KM classes, consulting with clients, researching and reading other practitioner and academic research, the author was confounded by the failures of all the practitioners, politicians, and academic actors to understand the importance of time and its complexities for strategic decision-making, managing knowledge or human capital, among many other aspects. Time is one of the hidden assumptions (dimensions) continually used without giving it a second thought. In practice, the misalignment of time horizons (present and future) and time frames of events constitutes an enormous impediment (both in planning/formulating and in implementa-

tion), but is rarely explicitly identified as an issue and/or studied (see a rare, very recent exception in Myllykoski [99]).

This point must be illustrated due to its importance. In the context of Knowledge Management for Sustainable Water Systems, the key players who take actions or make decisions, not only might have a diverse set of expertise, understating and knowledge of the subject at hand, but they also operate in a different time "space". Politicians' time horizon of the relevant future for their decision-making is different from that of the farmer, the hydrologist and the weather scientist. Their understanding of an event (and its time frame) and the implication the event might have within a complex system, including the impact of time-lags and complex feedback loops, could be startlingly different and diverse (from others) within the realm of their intentions, goals and their time horizon. This factor (of time) can explain by itself why knowledge is not managed effectively regarding important aspects of the sustainability of water systems. Corralling all the different actors into a single space of knowledge and coherent time for the purpose of advancing by fashioning effective decisions regarding Sustainable Water Systems can rarely happen if at all (See an illuminating and exceptional case of how visionary leadership was able to lead such a process successfully in Bilbao's Ria (Estuary of Bilbao), in [100].

To propose a solution to this gap (in academia) and a major issue in praxis, an updated model of the "garbage can model of decision-making" (Cohen et al. [101]) was proposed here, where time is a multi-dimensional construct having a synchronized (or not) time horizon, time frame, and event-time, or what I would define as the "quantum model of organizational time". This model advances the model of time described by Myllykoski [99] (building on Hernes [102]) in which she described the past and the future as a stream of events, that gets their "true" meaning at the present time, resulting from a stream of events, creating, or enabling a decision to be made, seeing time as agentic (p. 23). Events, or the bits of information perceived by the observer that are remarked by the actor as events, are seen as collapsing at the time of the decision. As such, it enables the freezing of the understanding of the past from the present perspective and planning for the future, confirming a present rational for the decision regarding the future [103]. Viewing time as agentic, and seeing the stream of bits of information coming from the past and going toward the future, collapsing at a time of a decision into one interpretation brings 'Schrödinger's cat' from the quantum realm into individual and organizational decision-making, and results in what I would call, the "individual quantum model of decision-making". Adding the complexity of multiple key actors with different time frames, etc. results in an "Organizational Quantum Model of Decision-Making", which is illustrated in Figure 4b. To increase the probability of a coherent and consistent time frame, this model is proposing the adoption of the framework of sustainable development and its 17 goals within the framework of circular economy, as mentioned above.

To conclude, the complexity of the reality of KM in Sustainable Development are overwhelming, as will be illustrated in the papers in this Special Issue. Such complexity is a result of the nature of Knowledge Management which could cut across ALL levels and units of analysis of the model, as well as across the unit of analysis in natural systems. Add to that the complexity of the diverse scientific areas and the diverse styles of learning and decision-making of the different actors, and you can see a complex networked system dealing with 'wicket problems' at its best.

### 2.2.3. Yin-Yang New/Sharing Knowledge Layer

We will next transition to the middle layer of the KM model, the Yin-Yang of new knowledge development and past/present knowledge sharing (Figure 5).

In the traditional academic literature, regardless if this is marketing literature, strategic literature, management literature and/or knowledge management literature there is a clear dichotomic distinction between the new and the old/present. Being that New Product Development versus Product management (in the marketing literature [104]); new venture [105], new entrants [106], blue ocean strategy [107] versus unsuccessful ventures,

industry rivalry, red ocean strategy (in strategic management literature). Or, leaders, vision new strategic plan versus managers, mission, operation management (in management literature [88]) and exploration, knowledge development versus exploitation, knowledge sharing (in the KM literature [73]). The model proposed here is suggesting that due to the arrival of the 4th Industrial Revolution (circa 2010 [75]) which is characterized by the fast pace of change in the environment, increased competition, higher dependence on complex supply chains, design thinking [108], technological enablers (e.g., 3d printing) among other factors, companies are no longer at a luxury to separately manage the two aspects. Today it is recommended for companies to be ambidextrous, agile, lean innovator, etc. For example, Russ et al. [109] reported that for the 65 companies that were studied, they almost found a perfect balance in their KM strategies between exploration and exploitation (49.6%), with a minimum of 10% and a high of 100% for exploration strategy. Companies can use internal networking as a medium to facilitate knowledge exchange to enable successful innovation processes that balance exploration and exploitation KM strategies [110]. Another example that identifies collaboration and sharing knowledge with suppliers early in the new product development is advantageous for the product developer [111]. Companies find that both continuous improvement and lean innovation are related to learning culture [112]. Learning was also found as a critical capability in achieving both short-term and long-term goals of strategic alliances (with product/market short term (sales/marketing) or long term (new product development goals) and technology transfer initiatives [113]. This may suggest that the appropriate question today is NOT what choice an actor should make (exploration versus exploitation) but how to effectively manage and/or efficiently manage ambidextrous processes, who will be performing them, where and when? See an example of an external support for exploration and internal support for exploitation in adoption of a new technology by Spanish hospital patients in a home setting, in Cegarra-Sánchez et al. [114]. Finally, going back to the time aspect mentioned above, one additional advantage of such an ambidextrous process is the enabling of a shared framework of reference and time collapse which should make the formulation and the implementation of strategies and or initiatives more coherent and consistent. A perspective that is at present badly needed when adopting sustainable [6] and circular economy framework.

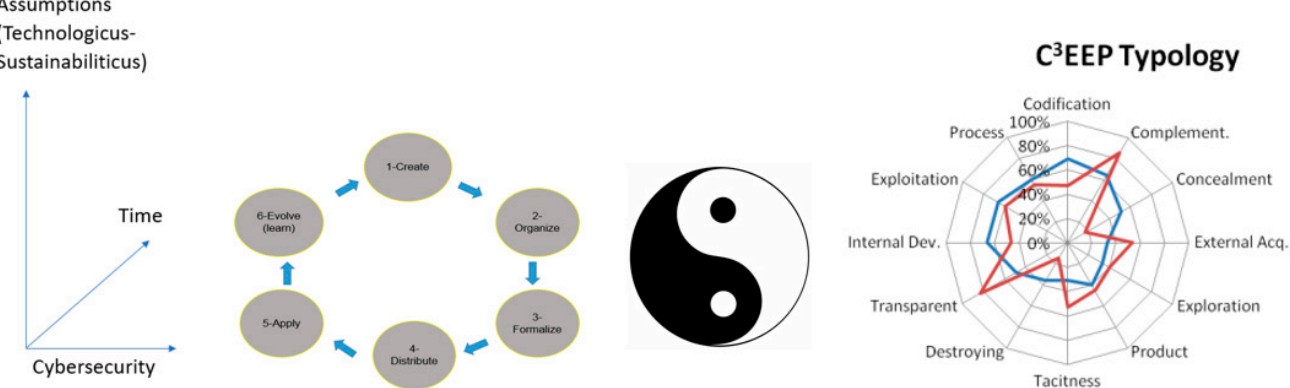

**Figure 5.** New knowledge developed and knowledge shared at the Knowledge/Human layer of Knowledge Management. Source: Author's slight modification of Russ [51] and Russ et al. [109].

Lastly, we will now transition to the Data/Digital Machine layer.

### 2.2.4. Data/Machine Layer

Early 2000s marks (see example [55]) the beginning of the Big Data area. Driven by cost advantages of commodity hardware and open-source software, increased investment in data systems, as a percent of IT spending on one hand, while on the other hand opening opportunities for innovative new business models and new insights that drive competitive advantages, which resulted in data stored (and collected) everywhere and in different

formats, which continues to grow exponentially (for example Internet of Things, smart factories, etc., [115]). What followed was a major development in data analytics tools and new generations of Artificial Intelligence models and tools that enhanced organizational knowledge management capabilities (see example [116]) and can support sustainable development (see examples at [117,118]).

Next, the four aspects of this layer will be covered: Learning, Decision Making, Systems and Human actors.

2.2.4.1. Learning

As mentioned earlier, learning is one of the most important processes in the context of our discussion here. Similar to an earlier discussion, a number of alternative models of learnings will be covered next. This discussion will start with the traditional 'space/object' of learning, the human learner. Then, the discussion will transition to the data/machine learning.

For over 20 years, during the author's teaching and public speaking, the author has suggested that the current business model of higher education is not sustainable. Teaching has remained the same over the last two thousand years, professor/instructor centered, knowledge pouring into 'empty' vessels. It seems that digital technologies, high costs, a corporate need for reskilling and upskilling, longer life expectancy [119–121] (and the pandemic) has finally caught up with this realty. MOOCs delivering certificate based, micro-credential courses and programs (e.g., Coursera, edX, Google, Udacity); skills based education (e.g., Global citizenship skills; Innovation and creativity skills; Technology skills; and Interpersonal skills), new models of learning (e.g., Personalized and self-paced learning; Accessible and inclusive learning; Problem-based and collaborative learning; and, Lifelong and student-driven learning) [122], among other new experiments and proposals are finally responding to growing educational and training needs, while leveraging the new technologies and the new models. One example will be discussed next.

The Open Learning model (see example at [123]) is enabling the student to choose and to study within their choice of a graphic view, aligning the view with their learning style (e.g., Kolb's styles mentioned earlier [85], p. 692). The model empowers the learner, by presenting them with the information about their learning to reflect on their skills, identify gaps in their skills, and plan their future learning [124]. This illustrates the opportunities that digitally embedded learning in a data/machine layer provides the interested learner of taking control over their own 'learning how to learn' [56]. Since learners needs change over time (e.g., skills become obsolete), such a tool is even more important for lifelong learners [125] as well as for companies interested in investing in their human capital.

The open learning model (see Figure 6a) is seen as one of alternative learners' analytic models, others being for example learning analytics dashboards and early warning systems [126].

On the other end of the spectrum of learning there are two mostly prominent and presently popular, successful, and widely implemented classes of pure AI/Digital learnings models, specifically Machine learning and Deep learning (for their application in economics and for other advanced methods see [127]; for application in agriculture see [128]; for application in logistics see [129]; for application in healthcare see [130]), which are now generally recognized as 'general purpose technologies' [131] (see Figure 6b).

Machine learning technology became a household name, when in 2016 the Google-owned artificial intelligence company DeepMind shocked the world by defeating Se-dol (The South Korean Go champion and presumably the best Go player in the world) four matches to one with its AlphaGo AI system. The games had a global impact, alerting the world to a new breed of machine learning programs [132].

Machine learning is a sub-field of artificial intelligence, and its structure mimics the neural network within the human brain. Machine learning models require human intervention to segment (label) data into categories in preparation for the process of training, which enables faster training, and is known as supervised learning. There are three types of learning tasks: Supervised learning, unsupervised learning, and reinforcement learning

(see for example [133]). Deep learning is a sub-field of machine learning, and its digital neural networks are structured based on layers of models, which can decode patterns within a given dataset. As such, deep learning algorithms do not require prior human intervention to segment data into categories, since it can leverage unsupervised learning to train itself. Early models of deep learning required high volume of training data to learn efficiently, while the more recent models can use a smaller data set [134]. The discussion above is summarized and modeled in Figure 6a,b.

Recent research found that AI and ML methods have been increasingly used to track progress toward some SDGs (mentioned above), however, the full potential of current state-of-the-art AI and ML methods and tools have not been fully utilized to enhance and accelerate their implementation [135].

### 2.2.4.2. Decision Making

Artificial Intelligence is playing a growing role in different aspects within organizations, especially in decision making roles. Expert opinions vary regarding the credence that AI currently plays in this area. Some consider AI as a decision support/augmentation tool for humans rather than as the automation of decision making to replace them [136]. Others, for example Iansiti and Lakhani [137] define the AI Factory as the scalable decision engine that operates at the core of the digital operating model of the firm that will revolutionize the landscape of business in the 21st century by providing an increasing rate of return on scale, scope, and learning [137] (p. 53). They see the algorithms and the data as guiding the core processes of the firm, marginalizing the role of humans in the critical path for value creation and delivery. Regardless, it seems obvious that more and more decisions will be taken by the AI algorithms and platforms [138,139], both vertically (context specific, see example at [140]) and horizontally.

Finally, the decision-making model (DMM) must accommodate human based DMM, Machine autonomous DMM and a mixed DMM (see Figure 6c). Simplistic models were presented by Colson [141]. A slightly modified version, with the addition of ethical constraints (the '*Homo Technologicus* versus the *Homo Sustainabiliticus*' dimension) can be seen in Figure 6b.

### 2.2.4.3. Systems

Cloud computing is undoubtedly a major driver in the rise of 'big data', by eliminating the need to maintain expensive computing hardware, dedicated space, and software [142]. The ability to store a larger quantity of data on smaller and cheaper physical devices as well as the increased speed of accessing and analyzing the data enabled value creation through the on-line synchronous analysis of high-velocity data streams at high speed data transfer which cloud computing could guarantee [143]. An example of a framework that makes big data practical is Hadoop's which brings distributed storage system and a data processing framework under one 'roof' [144].

Big Data can be defined (see seven definitions at [55], p. 10) along the following attributes/dimensions: 1. Volume—stands for the scale of data; 2. Velocity—denotes the analysis of streaming data 3. Variety—indicates different forms of data; 4. Veracity—implies the uncertainty of data; 5. Variability—refers to the complexity of data sets; 6. Validity—the quality of data is factually and logically sound; 7. Visibility—highlights that you need to have a full picture; 8. Verdict—the potential choice or decision made by a decision maker; and, 9. Value—is the data valuable, and will the information have the potential to create value to the decision maker (see more at [55], pp. 7–14).

Furthermore, the fusing of the physical and digital worlds resulting in a cyberphysical system [145,146] (for example the Intercloud and the Internet of Things-IoT) which still requires human interaction (e.g., learning, decision making, etc.), and transitioning to the up-and-coming fusing with the biological world [147] (biocyberphysical system) bringing us closer to the realization of *Homo Technologicus* and/or *Homo Sustainabiliticus* [6] is only mounting the prominence of data systems and data analytic techniques (e.g., mining). Such a growing networking effect and accelerated learning (by doing, by using, by interacting,

by observing [148], and by simulation [149]) are accelerating the adoption and reducing the cost of data systems, while accelerating the use of machine learning, data learning and overall, the scale and scope of data.

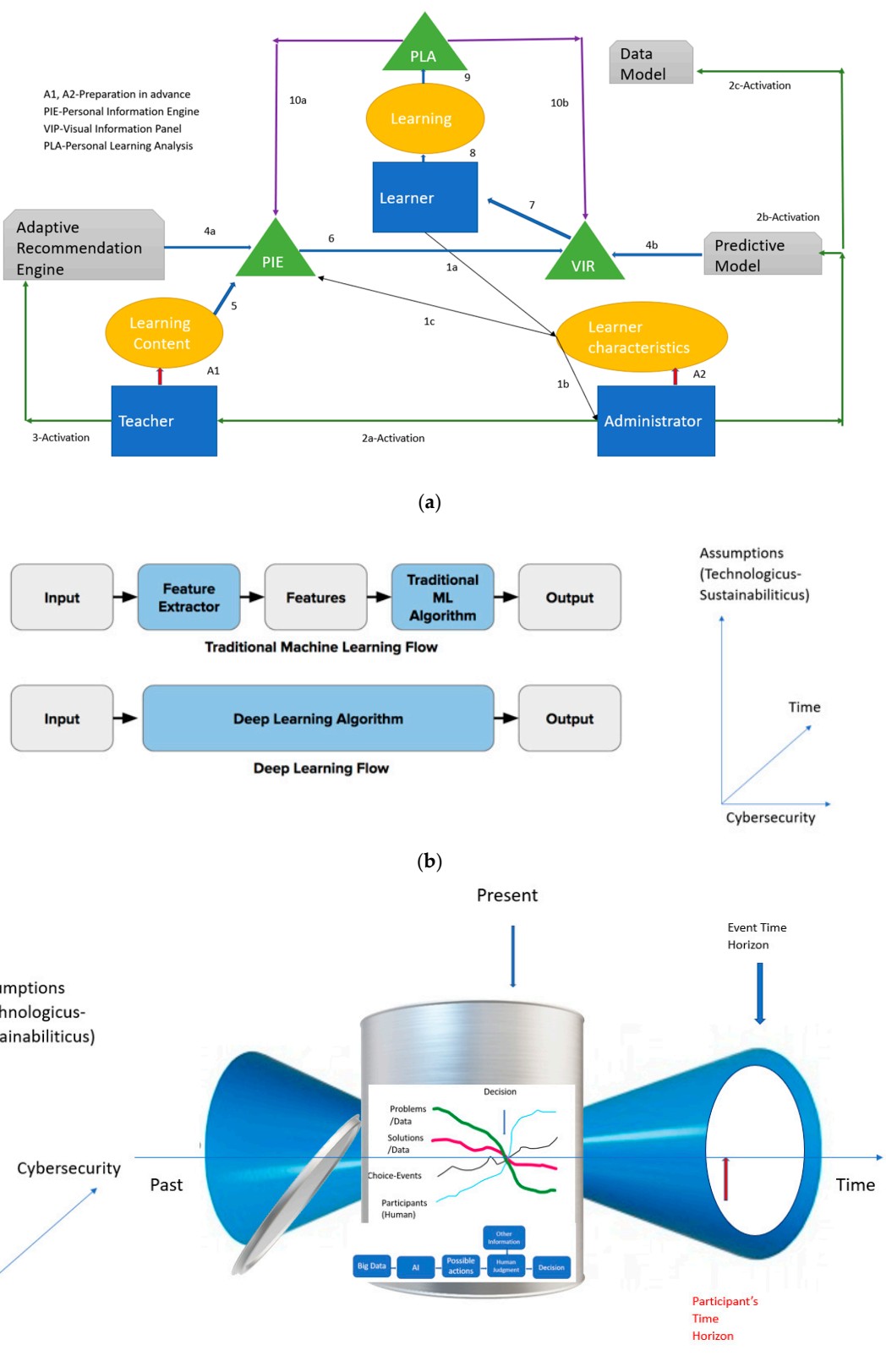

**Figure 6.** *Cont.*

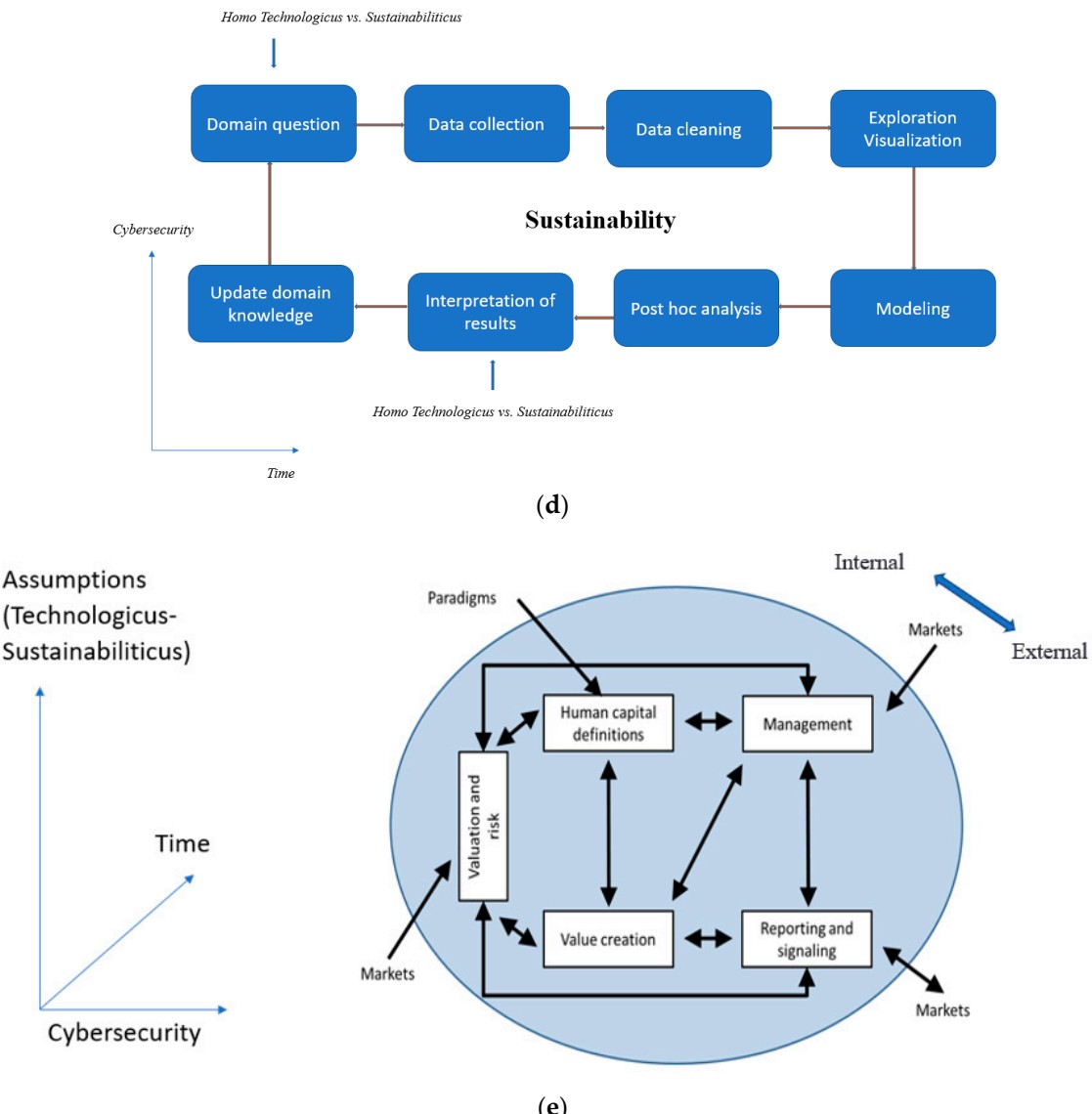

**Figure 6.** (**a**) The Open Learning model at the Data/Machine of Knowledge Management. Source: Author's modification of Liang & Hainan [56]. (**b**) The Machine Learning model at the Data/Machine of Knowledge Management. Source: Author's elaboration based on Kelleher [134]. (**c**) Quantum Organizational Decision-Making at the Data/Machine layer of Knowledge Management. Source: Author's slight modification of Russ [51]. (**d**) Data Science Life Cycle at the Data/Machine layer of Knowledge Management. Source: Author's slight modification of Yu and Kumbier [150]. (**e**) Human Capital at the Data/Machine layer of Knowledge Management. Source: Author's slight modification of Russ [51].

Next, the data system was modeled (see Figure 6d) using the data science life cycle suggested by Yu and Kumbier [150] with minor modifications; where stability is replaced by sustainability (triple bottom line) and the 'domain question' and the 'interpretation of results' modules is affected by ethics, values and norms of the entity, consistent with the *Homo Sustainabiliticus* perspective [6] (versus *Homo-Technologicus*).

2.2.4.4. Human Actors

Following the discussions above, probably one of the most interesting aspects of KM at the present time when AI and ML began to establish their dominance, is the 'space' that is still left for individual/human activity and the space and the nature of 'collaborative' activities of humans with/counter to AI/digital/robotics/machine actors. Here, we will focus on some aspects of those two 'spaces'. We will start with two subjects that illustrate

the symbiotic aspects of humans and digitalization, namely: (a) Leaders and AI, and (b) Citizen science and AI. This will allow us to open the scope of human activities from the very few in a business context to the many in a scientific context. Both of those anchors in our opinion, could and will play a major role in the future of sustainable development. We will close this part with the discussion of the role of tacit knowledge in human activity when we transition to data driven knowledge management.

What makes successful leaders in the context of Artificial Intelligence and Big data? Davenport and Foutty [151] suggested the following characteristics: quick learners of technology, set appropriate visions and goals (decision making), look for fast scaling based on data, and prepare the needed talent for the journey in a collaborative context [151]. Organizations driven by such leaders that install a culture orientation that is data driven (knowledge and decision making) have a significant higher probability to exceed their business goals [152]. To accomplish that, the leaders need to focus on data quality, identify which data is relevant for strategic decisions and develop the structure and talent for data management (separately from the IT systems management) [153]. Their understanding of technology and their implementation of machine learning must ensure that the use of data analytics is decision driven [154]. To increase the probability of their success while minimizing the risks, some new training tools are recommended (utilizing big data and AI), including, role playing [155] and simulation-based learning [156]. One of the reasons that could make the use of such tools a success in a complex context of this model as described here, is because it allows for the time horizons of the participating individuals and the time frame of the event they engage with to become coherent and consistent (to collapse, in terms used by the quantum model of DM), enabling an improved process of decision-making, as discussed above.

Juxtaposing leaders (internal) in the business context, one can envision citizen scientists (external), defined as a volunteer with varying levels of expertise engaged in scientific research or project [157]. Those volunteers can enhance the data collection capacity of the research team significantly, especially in terms of spatial and temporal scales and complement data collected by the cyberphysical network of sensors [157]. The volunteers can also be useful at the training stages of machine learning, codifying, labeling and monitoring the learning and the operation of the systems [158]. The volunteers also can complement the AI with serendipitous discoveries, increasing public engagement and in donating high in-kind resources [158]. Their involvement also incorporates specific risks and costs, for example, sampling biases, low accuracy and a need for training [158]. The involvement of citizen scientists can be especially beneficial for the social and environmental aspects of sustainable development due to the spatial and temporal scale and scope of data they can collect [158,159].

Next, we will discuss the tacit properties of KM. Humans are still better than AI with processing information that is fuzzy in nature, either regarding content-'know-what' (declarative knowledge), or process-'know-how' (procedural knowledge), or context (specifically strategic) and/or language (semantic and syntactic knowledge) and can find a way to clarify the confusion and/or to ask for additional input/information to learn the know-why (causal knowledge) [75]. Some of this knowledge is known as tacit (or implicit) knowledge, or a knowledge that we cannot verbalize or codify (explicit knowledge). Prawitz [160] suggested to differentiate between two different theses regarding tacit knowledge, a weak and a strong thesis. The weak thesis suggests that different actions or problems require different forms of knowledge (tacit or explicit). The strong one claims that tacit knowledge is always an ingredient in human action or an issue. He suggested that counterintuitively, the weak one might be a problem for AI, since it is related to the meaning of the language in use. The recent developments of machine learning suggest that he is probably wrong, and that AI is able to find an entirely new way for humans to solve an issue or devise an action, as long as the input and the outcome can be clearly defined. AI/machine learning does not need to "understand" an issue to resolve it, all that is needed is a 'path' or a pattern of getting from the input to the output, and data is the raw material needed for that [134]. This might suggest that the 'space' left for humans, at least in the near future, is the creative

space of creating new input and/or new outcomes. Which is why creativity, social skills, etc. are, and will be in demand, and the tacit knowledge supporting such activities is still in short supply [121]. Managing such tacit knowledge is still more art than science and leadership is one of those key areas where human involvement is of essence. One important aspect of leadership (and of management) is decision making (mentioned above). Here, it seems that more and more of the analytic decision making will be relegated to AI, while the intuitive decision making, at least for the near future, will still be a prerogative of the human decision maker [138].

Another 'space' where tacit knowledge is playing a critical role is in the public space, where tacit knowledge is embedded in the social milieu and includes cultural values and norms, or behaviors. Sanzogni et al. [161] suggested that at present, this is one of the weaknesses of the present AI generation. Recent research regarding (for example, racial or income) algorithmic biases [162,163], seems to support that claim. This subject will be discussed further under ethics below.

Finally, the human actors' talent (internal), was modeled (see Figure 6e) using the HC praxis model (Russ [96]) extended by using the life cycle of an AI-supported HR model suggested by Cappelli et al. [164], where the decision-making module is similar to the Colson [141] discussed above. This model is providing additional support to the understanding of the need to add decision making as an integral part of Knowledge Management by incorporating human capital/ talent management into it.

To summarize this part, we must analyze the impact of the synergistic effect of the data/machine layer with the human systems mentioned above in the context of sustainable development. One area that is of concern is the impact this layer has on jobs and the workforce. Clearly, some jobs are at a higher risk of disappearance (for example, jobs that are characterized by large data sets exist and well-defined patterns/functions from input to output) and the pace of the change will be affected by a number of economic (for example income elasticity and business process redesign) factors (see for example [131]). Unfortunately, what is missing from this discussion is the sustainability aspects, specifically, the social aspect (see discussion about ethics below) and the impact on the environment (or nature). It is questionable if free markets as presently structured will provide for a timely sustainable solution [6]. Some suggest that paradoxically, a planned economy might provide for an improved solution [165], while others consider an alternative model of currencies to be the only timely solution [166].

Number of propositions can be furthered at this point:

**Proposition 1.** *An effective use of KM will combine learning and decision making simultaneously utilizing the data and knowledge embedded in human talent and digital systems in a synergistic manner.*

**Proposition 2.** *Successful organizations will utilize an ambidextrous approach to advance new knowledge and to share concurrent knowledge consistent with their business goals and changing environmental conditions by using internal and external sources of human talent while creating a culture of accountability and push for challenging outcomes and providing space for learning and tolerance for failure simultaneously.*

**Proposition 3.** *Wicket and complex sustainable developmental issues will be resolved successfully by collaborating inter and intra organization agents that are able to create a shared and agreed time horizon for solutions based on incorporating interdisciplinary knowledge, involving the subjects in question, while incorporating triple loop learning to monitor progress.*

### 2.2.4.5. Additions

Earlier, we suggested that the dimension of time should be added to the model, as an overreaching aspect that should always be considered. Now, at closing, we are suggesting adding two additional aspects that should be always considered in the context of this model, namely: Ethics and Cybersecurity (see Figure 2).

2.2.5. Ethics

Abuses of data privacy and biases incorporated in algorithms haunted the early implementation of big data and machine learning in social media, politics, criminal justice and financial services (see example at [167]) among other implementations). Even academic research itself is not clear of some of those controversies (see example at [168]). This is forcing the data scientist and practitioners to engage in the difficult public discussions of legality [169], morals and ethics, which they avoided for quite some time by claiming that the technology is ethics-agnostic (see example at [170]). Additionally, the recent successful incorporation of AI and ML into medicine and healthcare with its long history of professional ethics and experience with dealing with ethical and moral issues, has forced the medical authorities to deal with ethical dilemmas related to AI and ML [171], similarly to other large-scale implementations that have broad public interest, like forcing the auto industry to deal with ethical and legal dilemmas when the autonomous cars had been 'put on the streets'. In the medical realm, the following four areas of ethical issues/concern have been identified as needed discussion in order to secure humanistic care for patients in a way that is consistent with the moral standing of the medical staff. The areas are: data stewardship (privacy, security, transparency, ownership), biases, safety in implementation-Nonmaleficence (and accountability), and societal implications (fairness, discrimination, liability) [172]. Research is suggesting that there should be expected frictions [173] or tradeoffs between different ethical dimensions, for example between transparency and accountability [174] or privacy and functionality [175]. The mentioned above ethical concerns could provide for a framework or an ecosystem for the ethical concerns in the context of smart systems (see example in Stahl and Wright [176].

In the context of this model, while considering the earlier part of this paper, which suggests we are probably in an infliction point in human history, we are proposing an alternative scope for the discussion of ethics, touching on the hidden assumptions behind many of the above-mentioned issues, as well as behind many legal aspects (not mentioned here, beyond the scope of this paper). I am referring here to the paradigms behind legal systems (discussed more in depth in Russ [6], specifically what is defined there as *Homo Technologicus* and *Homo Sustainabiliticus*. The definitions (modified from [6] are detailed below:

> *Homo-Technologicus*—"a symbiotic creature in which biology and technology intimately interact", so that what results is "not simply 'homo sapiens plus technology', but rather homo sapiens transformed by 'technology' into 'a new evolutionary unit, undergoing a new kind of evolution in a new environment'" (Longo [177], p. 23), driven by cost efficiencies and instrumental effectiveness within the techno-economic, universal and ontocentric perspectives and expecting adaptation of the 'homo sapiens' to the technology.

> *Homo sustainabiliticus*—a symbiotic being in which biology, technology and morality intimately interact driven by optimization and balance of costs of the technology solution while modifying it to optimize the user's adaptation, especially regarding her abilities and the social acceptance recognizing cultural and symbolic differences and environmental responsibilities based on biocentric ethics and the socio-philosophical point of view within her cultural, social, physical, logistic and legal context and cognizant of the ethical dilemmas of adapting the technology to her needs, specifically at the design stage.

For the purpose of this paper, there are three major differences between the two paradigms that are important to recognize (see Figure 7):

1. Technologies design and use for *Homo Sustainabiliticus* should be optimized for the effectiveness from the user/adaptor's perspective, and NOT (like in the case of *Homo Technologicus*) for the efficiencies (profit) from the technology creator/provider's perspective.

2.  Specifically, such design/use should provide the user/adaptor with "space" for using/adopting the technology within their values and morals, in the autopoietic meaning-self organizing "her context".

3.  For an effective design and use to happen, there is a need for transparency (especially at the design, development stage), the user must be educated appropriately to make educated choices about the potential tradeoffs (some of which were discussed above) and have the legal rights to do so, as long as they do not break other laws.

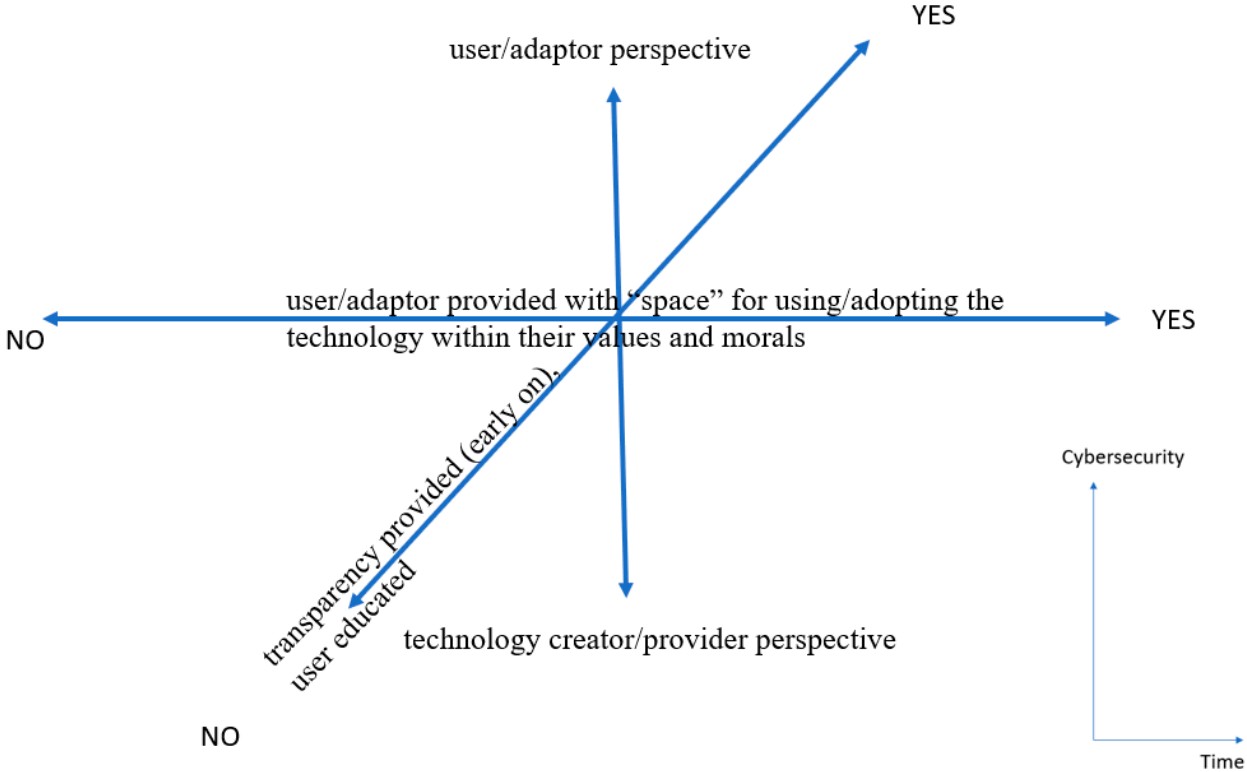

**Figure 7.** Three practical aspects of *Homo Technologicus* versus *Homo Sustainabiliticus.* Source: Author's elaborations

A number of propositions can be furthered at this point:

**Proposition 4.** *Effective technological solutions will utilize the Homo Sustainabiliticus paradigm, which will require subject education and room for technology modification for the unique subject characteristic. Such implementations will require additional resources and time.*

**Proposition 5.** *Efficient technological solutions will utilize the Homo Technologicus paradigm, which will require subject adaptation to the technology (or exclusion) regardless of their unique characteristics. Such implementations will require minimal resources and time.*

**Proposition 6.** *Regardless of the paradigm chosen, a human actor as decision-maker and as learner must be involved in the decision process at some capacity (even if only in the monitoring of the three feedback loops). Ethical concerns and security issues might cause issues that current autonomous systems are NOT truly ready yet to resolve (even if the current legal structure seems to approve).*

2.2.6. Cybersecurity

A reliable and secured system's infrastructure is critical in maintaining the knowledge management system and decision making needed to support the continuous and dynamic sustainable development as discussed in this paper. Cybersecurity risks regarding critical infrastructure, healthcare systems, financial systems among other systems, may result in

increased costs, reduced revenues, harm innovation and two-way communication between different constituencies so critically needed to support sustainable development. As such, cybersecurity must be an essential part of this model [178]. A recently proposed framework developed by the National Institute of Standards and Technology provides for principles and best practices for such risk assessment and also for its implementation within any organization [178]. Specifically, research is suggesting that deploying big data analytics can improve organization's cyber knowledge management capabilities, resulting in an organization's improved cyber agility [179]. One example of such application was recently proposed by Sarker et al. [180] and illustrates how building a dedicated (in this case security) data collection system and combining it with dynamic machine learning tools can improve on the five functions of the core structure of a cyber security framework mentioned earlier (identification, protection, detection, response, and recovery). Research is also pointing to new risks resulting from the implementations of ML and suggesting that almost by definition, while considering such risks, there are unavoidable tradeoffs between ML model's complexity, accuracy, and resilience that must be optimized for a specific context of their use (see for example [132]). In this model, we are recommending (as mentioned above) the use of the preferences as recommended by the Homo Sustainabiliticus paradigm. The results can be seen in Figure 8.

|  | Identify | Protect | Detect | Respond | Recover |
|---|---|---|---|---|---|
| Smart cybersecurity Systems & Services |  |  |  |  |  |
| Incremental learning & dynamism |  |  |  |  |  |
| Machine learning based security modeling |  |  |  |  |  |
| Security data preparing |  |  |  |  |  |
| Security data collecting |  |  |  |  |  |
| Cyber infrastructure |  |  |  |  |  |

Input from the *Homo Sustainabiliticus* paradigm →

**Figure 8.** Cyber Security. Source: Author's modification from NIST [178] and Sarker et al. [180].

*2.3. Managing Knowledge Boundaries*

As Cash et al. [63] suggested, boundary management is a critical success factor in any successful implementation of a knowledge system in a complexed and wicked problem in the context of sustainable development. Specifically, they recommended the creation of 'boundary objects' that are tangible results of cooperation between diverse constituencies to verify the mutual understanding and enabling successful implementation, within a dual accountability system. Such a system will provide for accountability with the 'mother' organization of the individual as well as within the collaborative structure in question. Tengo et al. [181] suggested a process that will structure the knowledge sharing between the diverse collaborators and their diverse knowledge base. By using (a) mobilization of the appropriate data and knowledge; (b) translating it, so it is usable for all participants; (c) negotiating and (d) synthesizing that data and knowledge to create a shared, usable, agreed and verified data and knowledge base; and (e) applying it into a form that is usable to the decision makers (or learners in our context as well); such complex and

multidisciplinary data and knowledge can be made relevant for timely and educated decisions, specifically in the context of managing knowledge boundaries. The proposed combined model of boundary management and learning in the context of this paper is presented in Figure 9, and a proposition can be added:

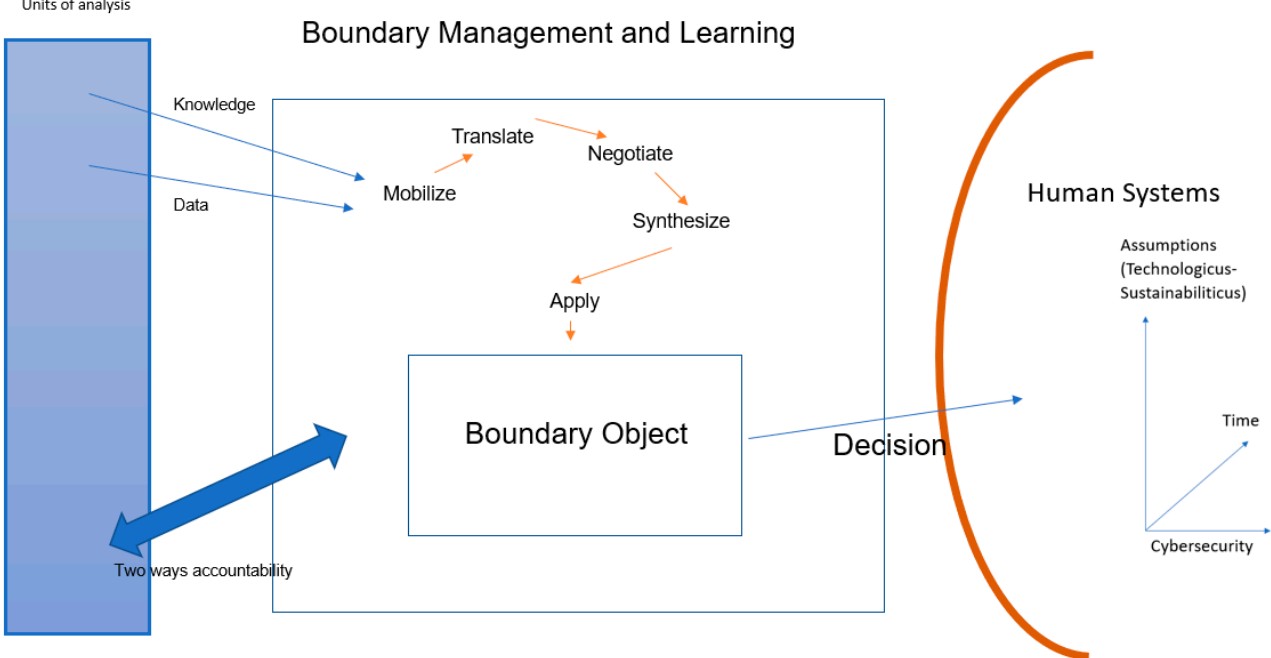

**Figure 9.** Boundary Learning and Management. Source: Author's elaborations based on Tengo et al. [170,181].

**Proposition 7.** *Boundaries between agents, ideas and/or knowledge-bases more effectively managed by learners (not doers) that are accountable both to their mother entity and to the collaborative structure when committed to a tangible object (product, outcome, etc.) and to an agreed and shared time frame. For such learning and delivery process to occur, a unique space (e.g., "ba") and additional resources and time will be required.*

## 3. Implications for the Future—The Macro Trends

### 3.1. The Macro Trends

This list of implications is based on and updated from [6,52,182–186].

Finally, the author is postulating with some macro trends resulting from the unique point in time of the history of the *Homo Sapiens* species as illustrated and discussed throughout this paper above.

This generation of *Homo Sapiens* might be the first civilization inventing its replacement (*Homo Technologicus* and/or *Homo Sustainabiliticus*) on this planet and beyond. As such, almost nothing (practically) learned from the past is relevant for the future.

Individuals will not be able to define themselves in terms of their jobs, since jobs will not be available to all who want them, unless government policies will drastically change (see an example in Hillenbrand and Money [187]).

Shrinking the half-life of knowledge will enable (or force) multiple careers in one lifetime (probably 4–8 careers), with about half of them not yet existing.

*Industrial economy*'s accounting, financial and economic indicators are of little relevance in the present and near future economy and using them to guide company strategy and/or governmental policies in many cases is causing more harm than good.

Trifurcation of the labor market is splitting the society into three (not two) segments: The 'have a lot', the 'have some', and the 'have none'. This suggests the death of the

average as a statistical and economic indicator and will require a different set of indicators (also different research methods by researchers).

Asking **new** questions/inquiries is the rare skill in high demand in the new economy. It can unlock new learning.

To define problems, one must ask questions and get reliable, relevant and timely answers/data. BUT, the first question to ask is: what is the appropriate unit of analysis and what are its useful features that are relevant to a solution for the issue at hand. Decision makers must learn to define issues better.

Delivering global/holistic (as a scope) answers will be the context for design thinking and fast implementation (lean innovation) including data-driven dataset design for successful solutions.

Knowledge is becoming a commodity; wisdom is less and less rare since humans in a 'collaborative and competitive' relationship with AI are using data to replace wisdom, by using deep learning, or open innovation, open systems, and crowd wisdom.

Education-traditionally seen as a social mechanism to accumulate knowledge and transfer it for future generations to be used, so the actors can linearly (and rarely discontinuously) improve on it, is less and less relevant in the new data driven economy. Alternative educational mechanisms should be (and are already starting) quickly developed.

Education, like health must be declared as a basic human right for all.

In a sustainable circular economy, health and education should be listed as an asset in the new 'smart accounting' system.

Life-Long-Learning (LLL) is here to stay (it is estimated that the half-life of knowledge of an Undergraduate degree at present is about 10 years) and must be embedded in a daily routine for those individuals and organizations who want to actively participate in the economic (for profit) or social (not for profit) life. At least a living income must be secured for the rest.

The 'have jobs' and 'have no jobs', will be only one divider within societies. Other divides will include have and have not access to broadband or own a smartphone, connected to the Internet of Things and Social Media or withdraw from, among others. To minimize the social costs of such divisions, access to digital networks must be also declared as a basic human right.

Global environmental and climate (or pandemics) issues cannot be solved by using national governments as a vehicle for a solution. Digital cooperation and collaboration between different actors on a global scale might be better in tackling such and other global wicked problems (see examples at [188,189]). National legislation, policies and politics should avoid staying in the way.

The current fiat currency and monetary system at large is not an appropriate solution to many of the issues listed above. Crypto-currencies (like Bitcoin) and national digital currency are just the first generation of the alternative solution.

Morality and ethical concerns and frameworks (see example at [190]) as expressed by the general public by using digitally trusted shared ledgers will have to replace the badly lagging behind and twisted by private interest (national) legislation, to have an effective (to ensure evolutionary variety/diversity) impact on a technological development path, if *Homo Sustainabiliticus* is to survive.

Similar platforms should be used to resolve the build-in ethical paradoxes regarding the tradeoffs between competing sustainable development goals (for example, Equity and Natural Capital Stock, see [191]).

*3.2. Implications for Research-Theory Building and Model Testing*

The framework (Figure 2) and the models (Figures 3–9) proposed, and propositions suggested throughout the paper could provide a fertile ground for a number of (resulting from the broad scope of the subjects covered) middle-range theories, and for specific models and propositions to be tested.

Such theories and model testing could be conducted initially using carefully selected case studies and/or rigorous literature reviews, resulting in specific propositions (see example in [192] where socially driven entrepreneurs respond to opportunities in different urban units of analysis, while collaborating (sharing knowledge) with diverse stakeholders and developing new ventures).

Another contribution of the paper can be in unifying diverse sets of sciences and or theories and models. Clearly, the well documented failures of mitigating environmental disasters, water scarcity among others (e.g., [193]) indicates (at least partially) the need for an improved understanding and fast learning from such failures that cannot be wasted. A recent discussion about the "new science" of Socio-hydrology and its relationship with the more tradition hydrology [194] and the need to incorporate human activities as part of co-evolving systems, is clearly lacking the incorporation of the psychological needs of the individual political decision maker, their learning capacity, their time horizon and their access to current and newly developed knowledge. Such ongoing failures only accelerate the need for such an encompassing, cross and multi-disciplinary theories.

### 3.3. Implications for Policy Making

For policy makers, the most important implications are ensuing from the quantum model of decision making, and the boundary management and learning models. The quantum model of decision making is suggesting that agreeing on a shared agenda within an agreed (explicitly shared and committed too) time horizon is a critical aspect of an effective decision-making process. To achieve a shared vision of the future, the model is sanctioning the bringing forward (to the time the decision is taken) of the appropriate past, as needed for such an agenda. To arrive to such a future, in the case of a complex and wicked sustainable development issue, learners must be brought to the decision-making 'table' to learn and to manage the process of providing the data and knowledge needed for effective decision making and to avoid (what might seem) efficient processes that will in all probability result in unexpected consequences.

The rest of the models and the framework could be helpful at different points of processes, but the two listed above are crucial for successful sustainable development policies.

### 3.4. Implications for Practitioners

Practitioners should find the models (above and beyond those listed above for policy makers) useful for different applications and processes. Due to the broad scope of areas the models can be applied to, it will be pretentious to provide specific advice above and beyond a small number of implications, some of which are listed below.

Practitioners, both doers and thinkers/learners (who are motivated by a diverse set of motivators and value different outcomes (e.g., [195]) must collaborate (learn and decide) seamlessly for an organization to be effective, efficient, and agile (see example at Santini et al. [196]).

A decision-making process must be data (reality) based utilizing data and knowledge, human talent and AI in tandem. Decisions must be implemented in a timely manner (design thinking could be helpful here), keeping in mind that data and knowledge are one of the expected outcomes of any implementation at any stage. One of the drivers of the implementation should be scalability, while keeping in mind the sustainability mind set (e.g., triple bottom line or multiple constituencies).

## 4. Conclusions

In conclusion, the changing *Padkos* environment has created a new complex ecosystem, unknowable and confronting the *Homo Sapiens* on a global scale as individuals and as societies, one that would question and challenge almost every assumption that was taken for granted and accepted in the natural, social, economic and political space. Two contradicting options for dominant paradigms that could frame the future, namely, *Homo Technologicus* and *Home Sustainabiliticus* were articulated. In this context, a three-layered

model of knowledge management was proposed, to facilitate learning and decision making of agents at any social, digital or physical level of aggregation. The new environment seems to impose an integrated learning and decision-making processes of new and standing knowledge, enabling the simultaneous fusion of data and knowledge, man and machine as a preferred model of self-organizing autopoietic system. For such a system, to make sustainable and humanistic choices, a quantum model of decision making was proposed, one that explicitly articulates the collapse of the envisioned future and the chosen past at the time a decision is taken with a specific time horizon in mind. For individuals and organizations to make such an educated decision as an autopoietic system, a boundary management and learning model was proposed, encapsulated in an ethical dilemma framework of the dominant paradigm. If the choice between such paradigms is still an option, decisions need to take place in the immediate future.

**Author Contributions:** The author has read and agreed to the published version of the manuscript.

**Funding:** This research received no external funding.

**Institutional Review Board Statement:** Not applicable.

**Informed Consent Statement:** Not applicable.

**Data Availability Statement:** No new data were created or analyzed in this study. Data sharing is not applicable to this article.

**Acknowledgments:** The author is grateful to Sara Scipioni for her help with editing this paper and to the three anonymous reviewers for their helpful comments.

**Conflicts of Interest:** The author declares no conflict of interest in the collection, analyses, or interpretation of data; in the writing of the manuscript, or in the decision to publish the results.

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
