# Peer review of "Knowledge Management for Sustainable Development in the Era of Continuously Accelerating Technological Revolutions: A Framework and Models"

_sustainability, doi:10.3390/su13063353_

Round 1
Reviewer 1 Report
The paper is written from an interdisciplinary point of view on the topic of the phenomenon of continuously accelerating technological revolutions. The paper considers different points of view of a new human society era. It first deals with the challenges for humanity brought by new technological revolutions, then analyzes the existing models of knowledge management from the angle of sustainable development and proposes their improvement and setting up new models. The three layer model of knowledge management and boundary management and learning model specific to knowledge sharing are especially promoted. At the end of the paper, a discussion was presented regarding several implications for the future period, ie. megatrends that will be significant for human society in the future.
The article is well written and analyzes is presented appropriately. The research question is original and the results are interpreted appropriately. Nevertheless, I am of the opinion that the hypotheses of the paper should be more clearly emphasized in the text itself and put in abstract as well. Next, I would suggest changing the title because I find it too long. Chapter 3 should be reformulated into two separate chapters that make up the discussion and conclusion, without writing in the first person singular.
The English language is appropriate and understandable. However, does not explain enough abbreviations that are in it. Namely, each abbreviation must be explained when it is used for the first time, which is not the case in the paper. The abbreviation „Padkos“ should be better explained and unique, as it appears in different variants like „Pankos“ and „Patkos“. Also, all figures should be mentioned in the text and may need to be numbered more adequately.
Author Response
Reviewer 1 |
Response |
The article is well written and analyzes is presented appropriately. The research question is original and the results are interpreted appropriately. |
Thanks to the 3 reviewers for their kind words and wise and helpful advice. |
Nevertheless, I am of the opinion that the hypotheses of the paper should be more clearly emphasized in the text itself and put in abstract as well. |
Seven propositions were added; see [751-765 ] [841-855 ] and [906-911] and also mentioned in the abstract and introduction. |
Next, I would suggest changing the title because I find it too long. |
Title was changed and shortened. |
Chapter 3 should be reformulated into two separate chapters that make up the discussion and conclusion, |
Chapter 3 now includes the two requested parts plus a few additions, see [915-1064]. |
without writing in the first person singular. |
Language changed throughout the paper as suggested. |
However, does not explain enough abbreviations that are in it. Namely, each abbreviation must be explained when it is used for the first time, which is not the case in the paper. The abbreviation „Padkos“ should be better explained and unique, as it appears in different variants like „Pankos“ and „Patkos“. |
Abbreviations corrected and explained. |
Also, all figures should be mentioned in the text and may need to be numbered more adequately. |
All figures are mentioned in the text, see [340-344], 568; Figure numbers were retained to be consistent with the written content. |
Reviewer 2 Report
This article suggests a current and attractive topic for the academy. The research is timely and worthwhile. The research problem is clearly defined. The authors provide fresh insight into the field.
The work structure is excellent and well-articulated. The literature review is detailed and thorough. To review the scientific sources from the industry can offer work of his colleagues: Korobiichuk I., Fedushko S., Juś A., Syerov Y. Methods of Determining Information Support of Web Community User Personal Data Verification System. In: Szewczyk R., Zieliński C., Kaliczyńska M. (eds) Automation 2017. ICA 2017. Advances in Intelligent Systems and Computing. Springer International Publishing, 2017. Volume 550. pp 144-150.
The results are explained in a clear and detailed manner.
Congratulations on a job well done.
Author Response
Reviewer 2 |
|
To review the scientific sources from the industry can offer work of his colleagues: Korobiichuk I., Fedushko S., JuĹ› A., Syerov Y., 2017. |
Incorporated-see [326-331]. |
Reviewer 3 Report
The paper provides interesting insights on the thematic and incorporates the theoretical and empirical implication of the most recent technological advancement concerning knowledge management (KM) in the context of sustainable development as an updated conceptual model. The paper is well written and the arguments are presented in a logical and coherent manner.
I recommend that the paper is accepted for publication after minion revisions, mostly related to the introduction section, namely:
- The introduction section does not provide sufficient information on the methodological approach used to examine the research problem.
- Please also consider briefly explain the structure and organization of the paper.
Line 107- financial crisis instead of final? crisis -consider revising
Best wishes,
The reviewer
Author Response
Reviewer 3 |
|
The introduction section does not provide sufficient information on the methodological approach used to examine the research problem. |
The methodology was illustrated by an example, [200-211]. A more rigorous methodology was recommended for the next stage of theory development using the model as a building block (in research implications). |
Please also consider briefly explain the structure and organization of the paper. |
Structure added in section 1.1. ; [68-94]. |
Line 107- financial crisis instead of final? crisis -consider revising |
The paper was proofread again. |